# Selective hydrogenolysis of catechyl lignin into propenylcatechol over an atomically dispersed ruthenium catalyst

Shuizhong Wang[1], Kaili Zhang[1], Helong Li [1], Ling-Ping Xiao [2] & Guoyong Song [1✉]

C-lignin is a homo-biopolymer, being made up of caffeyl alcohol exclusively. There is significant interest in developing efficient and selective catalyst for depolymerization of C-lignin, as it represents an ideal feedstock for producing catechol derivatives. Here we report an atomically dispersed Ru catalyst, which can serve as an efficient catalyst for the hydrogenolysis of C-lignin via the cleavage of C−O bonds in benzodioxane linkages, giving catechols in high yields with TONs up to 345. A unique selectivity to propenylcatechol (77%) is obtained, which is otherwise hard to achieve, because this catalyst is capable of hydrogenolysis rather than hydrogenation. This catalyst also demonstrates good reusability in C-lignin depolymerization. Detailed investigations by model compounds concluded that the pathways involving dehydration and/or dehydrogenation reactions are incompatible routes; we deduced that caffeyl alcohol generated via concurrent C−O bonds cleavage of benzodioxane unit may act as an intermediate in the C-lignin hydrogenolysis. Current demonstration validates that atomically dispersed metals can not only catalyze small molecules reactions, but also drive the transformation of abundant and renewable biopolymer.

[1] Beijing Advanced Innovation Center for Tree Breeding by Molecular Design, Beijing Key Laboratory of Lignocellulosic Chemistry, Beijing Forestry University, Beijing 100083, P.R. China. [2] Center for Lignocellulosic Chemistry and Biomaterials, Dalian Polytechnic University, Dalian 116034, P.R. China. ✉email: songg@bjfu.edu.cn

Lignin is a major component of nonedible biomass and represents one of the few renewable aromatic biopolymers[1,2]. The distinctive aromatic backbone of lignin makes it unique among biopolymers and potentially valuable as a feedstock for aromatic chemicals[3–5]. Among possible approaches for the depolymerization of lignin[6–10], hydrogenolysis of lignin (reductive method) by heterogeneous metal catalysts remains one of the most promising methods for producing phenolic monomers in high yield[11–19]. However, catalytic hydrogenolysis of lignin into aromatics in a unitary and purifiable manner is currently problematic[7], as most lignins derived from woods or grasses are composed of several subunits with complex and recalcitrant structures[1,2].

Very recently, an unusual catechyl lignin (C-lignin) was found in a series of seed coats of *Vanilla planifolia*[20,21], *Melocactus obtusipetalus*[22], and *Ricinus communis* (castor) (Fig. 1)[21]. Such lignin is shown to be essentially a homopolymer biosynthesized from caffeyl alcohol owing to the lack of O-methyltransferase[21,23]. Benzodioxanes are identified as the dominant linkages. The C-lignin displays stability toward harsh (i.e., Klason and acidic LiBr procedures) or mild (i.e., enzymic treatment) biomass pretreatment conditions, so that it can be fractionated from raw seed coats without condensation[24,25]. These features of C-lignin render it an ideal lignin archetype for valorization[25]. As the only subunit in C-lignin, the catechol skeleton occurs in a variety of natural products, pharmaceuticals, and functional materials[26,27]. Chemical synthesis of catechol derivatives usually requires lengthy steps and harsh reaction conditions, with low selectivity and conversion rates[28]. The depolymerization of C-lignin, in theory, can produce catechol derivatives in a unitary fashion;[29] however, there existed a mere handful of literature examples (Fig. 1)[25,30,31]. Ralph and coworkers described the hydrogenolysis of C-lignin derived from vanilla seed coats by commercial Pt/C, Pd/C, and Ru/C catalysts, from which catechol monomers bearing propyl and/or propanol end-chains were generated up to 89% yield (based on the molar amount of caffeyl alcohol)[25]. Beckham and Román-Leshkov used Ni/C to treat vanilla seed coats directly, and C-lignin component was depolymerized into a series of catechol monomers with 7.9% yields (based on the weight of vanilla seed coats)[30]. An early report

by D'Souza also found that a mixture of catechols could be generated from hydrogenolysis of candlenut lignin by a Cu-PMO (PMO = porous metal oxide) catalyst[31].

Despite these advances, several critical limitations remained to depolymerize C-lignin in a practical version. First, the utilization efficiency of precious metals in supported catalysts is far less than the requirement for industrial applications. For example, the use of Pt/C, Pd/C, and Ru/C catalysts gave turnover numbers (TONs) in the order of $4$ $\text{mol}_{\text{catechols}}$ $\text{mol}_{\text{metal}}^{-1}$ (Fig. 1)[25]. Second, catechols with an unsaturated end-chain (such as caffeyl alcohol and propenylcatechol) are reason to be more valuable lignin-derived platform molecules because they can be functionalized diversely. However, the unsaturated catechols are hard to reserve under commercial catalysts because of their high hydrogenation affinity[30]. In this context, engineering a catalyst, that can steer the cleavage of two C—O bonds in benzodioxanes efficiently, together with circumventing the hydrogenation of C=C bonds selectively, is highly desired but challenging.

Recent experimental results implied that atomic dispersion of metal species including clusters and single atom, can dramatically improve the atomic economy of catalysts as well as catalytic activity and/or selectivity[32–34]. These catalysts have been proven to be effective in various reactions of small molecules, such as CO oxidation[35,36], methane oxidation[37], selective hydrogenation[33,38–40], ammonia synthesis[41], hydrodeoxygenation[42], three-way catalysis reactions[43], and so on. Applying the atomically dispersed metal catalysts in macromolecule transformation, for example, the depolymerization of lignin, has not been reported yet.

As hydrogenolysis of the C—O bonds is metal-dependent, depolymerization of benzodioxane structures in C-lignin may likely be accomplished by the choice of a metal center having a strong activity to hydrogenolysis C—O bonds, such as Ru[11,15,19]. In this work, we reported a Ru/ZnO/C catalyst prepared by loading Ru in situ synthesis of a metal-organic framework (MOF) and subsequent pyrolysis[39,44], where the Ru is fabricated as either clusters or single atom on porous carbon. This catalyst exhibits high activity towards hydrogenolysis of C-lignin derived from castor seed coats, thus giving catechol monomers in high yields with high TONs. Furthermore, high selectivity towards propenyl-substituted catechol

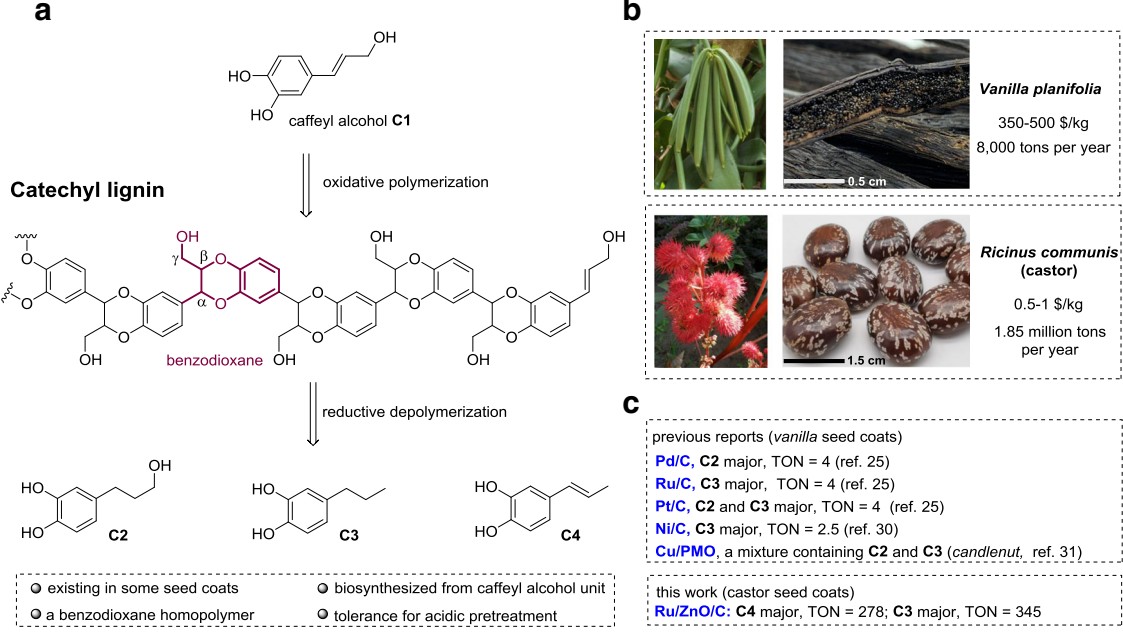

**Fig. 1 Catechyl lignin (C-lignin) structure and characteristics. a** Overview of C-lignin structure. **b** Comparison of price and annual output between *Vanilla planifolia* and *Ricinus communis* (castor). **c** Previous reports on C-lignin depolymerization.

was achieved, because the C−O bonds cleavage is more efficient than the C=C bonds hydrogenation with Ru/ZnO/C catalyst. The current catalyst is stable, and can be cycled up to six times without significant loss of performance. On the basis of the reactivity model compounds and related catechols, the plausible pathways for Ru/ZnO/C-catalyzed hydrogenolysis of C-lignin were also proposed.

## Results

**Preparation and characterization of catalyst.** Loading Ru species took place in situ along with the preparation of Zn-BTC MOF[45] (BTC = 1,3,5-benzenetricarboxylic acid) by merging $Zn(OAc)_2$ and BTC with trace $RuCl_3·3H_2O$ in one pot. Subsequent thermal treatment at 750 °C under $N_2$ afforded Ru/ZnO/C catalyst. The weight percentage of Ru element was determined as 0.2 wt% according to the analysis of inductively coupled plasma-atomic emission (ICP-AES) (Supplementary Table 1). Energy dispersive spectroscopy (EDS) spectra indicated the presence of Ru, Zn, C and O elements. The X-ray diffraction (XRD) patterns of Ru/ZnO/C showed typical peaks for wurtzite ZnO phase (JCPSD No. 36-1451);[46] no characteristic peaks for Ru were observed in XRD, probably because of low loading and/or high dispersion of Ru element (Supplementary Fig. 1). In X-ray photoelectron spectroscopy (XPS) spectra, two signals at 1044.5 and 1021.2 eV in the Zn 2*p* spectrum were assigned to ZnO specie (Supplementary Fig. 2)[46]. The image of aberration-corrected high-angle annular dark-field scanning transmission electron microscope (HAADF-STEM) revealed that the brighter spots assigned to the heavier Ru atoms existed as either clusters (red circle) or single atoms (blue circle) on the carbon support, with no observation of nanoparticles (Fig. 2a, b). The high dispersed manner of Ru was also confirmed by the mapping of elemental mapping studies (Fig. 2c).

To further understand the structure of Ru in Ru/ZnO/C, we performed X-ray absorption near-edge structure (XANES), extended X-ray absorption fine structure (EXAFS) spectroscopies

and a wavelet transform (WT) analysis. The Ru K-edge XANES curves displayed that the energy absorption threshold value of Ru/ZnO/C was higher than that of elemental Ru and lower than that of $RuO_2$, indicating the Ru species feature positive charge (Fig. 2d)[39,41]. In EXAFS spectra, two intense peaks resonated at 1.52 and 2.25 Å were noticed, which can be ascribed to Ru−O and Ru−Ru scattering, respectively (Fig. 2e)[39,41]. Quantitative EXAFS analyses were carried out, and the fitting results are displayed in Supplementary Table 2 and Supplementary Fig. 4. The WT contour plots showed intensity maximums at 3.1 Å and 7.9 Å, relating to Ru−O and Ru−Ru contributions, respectively (Fig. 2f; Supplementary Fig. 5)[39]. These results suggested that Ru displays in the form of both single atom and cluster in Ru/ZnO/C, being in line with the observation in STEM.

**Catalytic hydrogenolysis of C-lignin.** Castor (*R. communis*) seeds have been widely used for the production of nonedible oils[47]. Pioneering reports indicated that the castor seed coats are enriched in C-lignin biopolymer[21,48], which made it a plentiful and inexpensive C-lignin feedstock (0.5−1 $/kg) instead of vanilla seeds (350−500 $/kg) (Fig. 1). Lignin samples, derived from epicarp and endocarp of castor seeds (Supplementary Fig. 6), were isolated through the combination of enzymatic and mild acidolysis treatment[17]. Thioacidolysis and 2D HSQC NMR analysis clearly revealed that the epicarp lignin presented a typical herbaceous lignin structure, where guaiacyl (G), syringyl (S), and *p*-coumaric acid (*p*CA) moieties, as well as β-O-4 linkages were detected (Supplementary Figs. 7 and 8). In contrast, thioacidolysis of isolated endocarp lignin only released caffeyl alcohol monomers with essentially no G and S monomers (Supplementary Fig. 8), indicating it was a typical C-lignin. 2D HSQC NMR spectroscopy of endocarp C-lignin showed that benzodioxane structures were the predominant linkages, accounting for over 86% of total detectable dimeric units (Fig. 3e). Resinol species were also observed, whereas

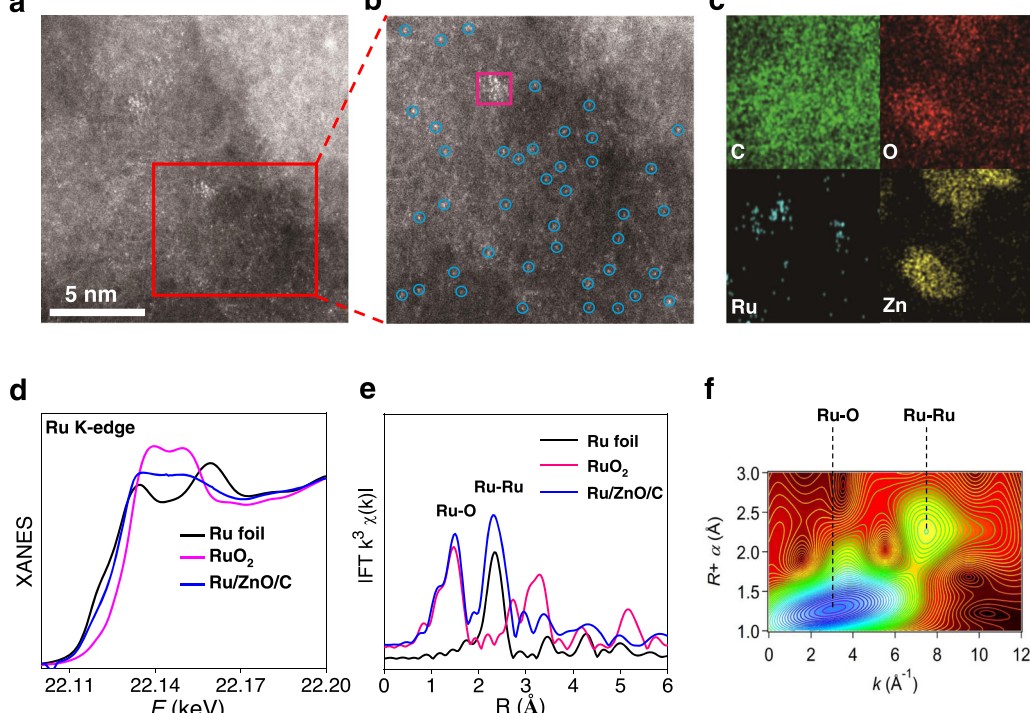

**Fig. 2 Characterizations of the Ru/ZnO/C catalyst. a, b** HAADF-STEM images. **c** STEM-EDS elemental mapping results. **d** Ru K-edge XANES spectra. **e** EXAFS Fourier transformed (FT) $k^2$-weighted $\chi(k)$-function for Ru K-edge. **f** Wavelet transforms for the $k^2$-weighted Ru K-edge EXAFS signals in Ru/ZnO/C.

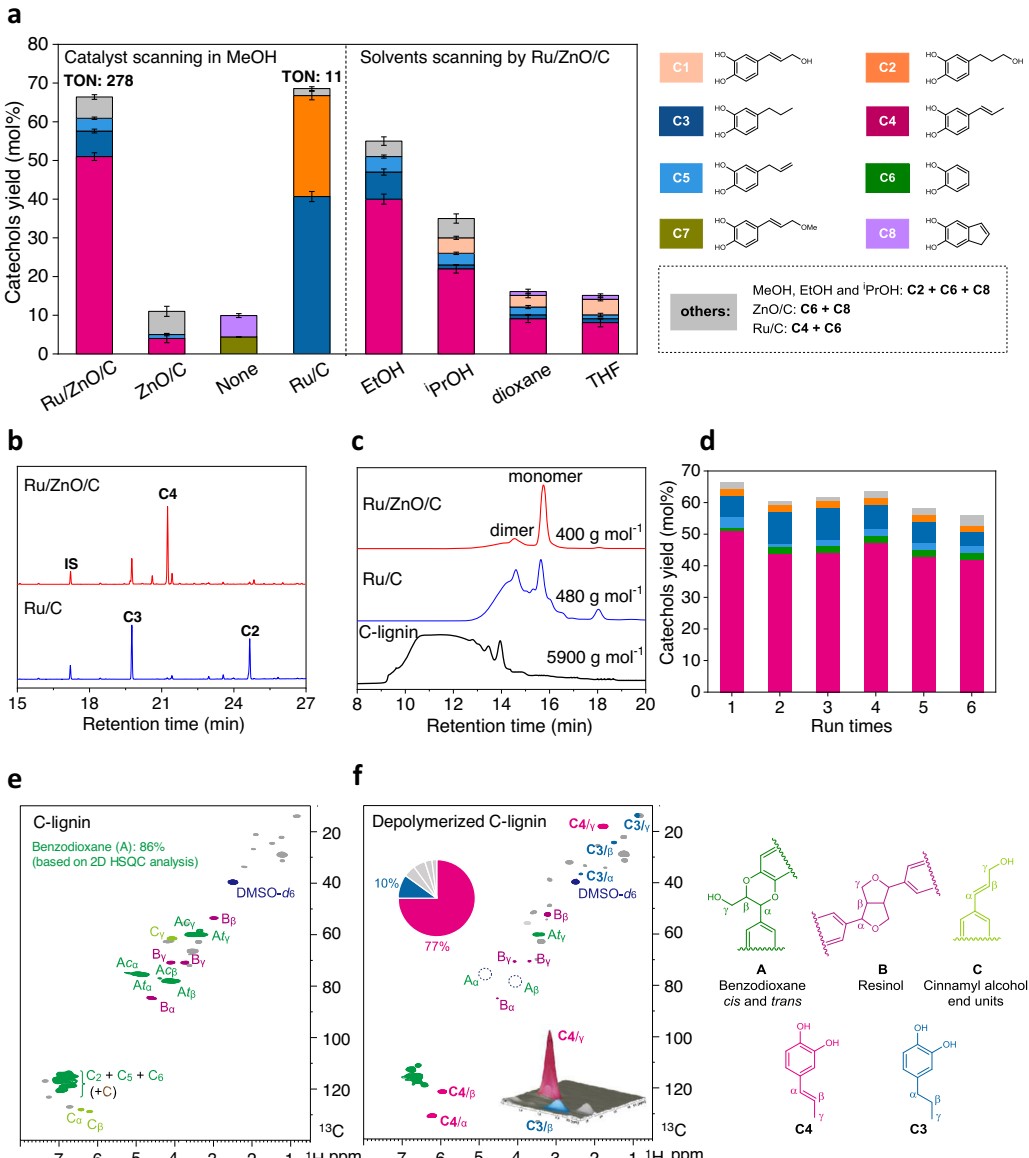

**Fig. 3 Hydrogenolysis of C-lignin over Ru/ZnO/C catalyst. a** Comparison of catechols yield and selectivity of different catalysts and solvents, and corresponding products distributions. Reaction conditions: C-lignin (50 mg), catalyst (15 mg, 30 wt%), solvent (10 mL), 200 °C, 3 MPa $H_2$, and 4 h. TON means turnover numbers. (Error bars = standard deviation; $n = 2$; for Ru/ZnO/C, $n = 3$). **b** Comparison of the catechols products (trimethylsilyl) derivatives) after the Ru/ZnO/C and Ru/C catalyzed processing of C-lignin by GC spectra. **c** Molecular weight distribution of C-lignin and oily products from catalytic reactions via GPC. **d** The reusability of Ru/ZnO/C catalyst. **e** 2D NMR spectrum of C-lignin isolated from castor seed coats (DMSO-$d_6$). **f** 2D NMR spectrum of oily product from Ru/ZnO/C-catalyzed reaction (DMSO-$d_6$), and corresponding structural distributions of C-lignin and main products.

β-O-4 structures were not detected. Biomass compositional analysis of endocarp C-lignin gave a Klason lignin content value as 83.1% (Supplementary Table 3); however, this content may be overestimated because some acid-survived lipids and proteins in seeds would be considered as "Klason lignin"[30]. We performed quantitative ¹³C NMR spectroscopy[25], from which the molar concentration of caffeyl alcohol units in endocarp C-lignin was calculated as 2.49 mmol g⁻¹ (Supplementary Fig. 9). This result suggested that the actual content of caffeyl alcohol units in current C-lignin sample is 41% by weight.

We then chose the C-lignin sample isolated from endocarp of castor seeds to test the catalytic performance of Ru/ZnO/C. The hydrogenolysis of C-lignin was initially carried out at 200 °C and 3 MPa of $H_2$ in MeOH for 4 h in a batch, by using 30 wt% of Ru/ZnO/C as a catalyst. This reaction led to a soluble fraction (70 wt%) and solid residue (21 wt%), with 91% mass balance

based on the weight of the original C-lignin sample. Analysis by gel-permeation chromatography of the soluble fraction revealed a significant decrease in molecular mass (400 g mol⁻¹) relative to the initial C-lignin biopolymer (5900 g mol⁻¹). A dominant peak corresponding to the monomeric catechols was generated after catalytic reaction (Fig. 3c; Supplementary Fig. 10). To identify and quantify the monomers, the soluble fraction was silylated with N,O-bis(trimethylsilyl)trifluoroacetamide (BSTFA) and was analyzed on gas chromatography-mass spectrometry (GCMS) by comparison with the authentic samples. Overall, 66% yield of well-defined catechol monomers was normalized based on the total molar concentration of caffeyl alcohol units in C-lignin (Fig. 3a). The TON, was calculated as 278 mol_catechols mol_Ru⁻¹ based on the total number of moles of Ru in the catalyst. The detailed distribution of catechol monomers is depicted in Fig. 3b and Supplementary Table 4. Propenylcatechol **C4** was identified

as the major depolymerized product in 51% yield with 77% selectivity. This product was proposed to be formed via the cleavage of dual C−O bonds in benzodioxane structure and the loss of the hydroxyl group at γ-position. Owing to the high selectivity, C4 could be isolated in a pure fashion (18 wt% or 48 mol% yield based on the C-lignin sample) by chromatographic column (Supplementary Fig. 11). Other catechyl monomers bearing different end-chains at para-position, such as propylcatechol C3 (6.6%), allylcatechol C5 (3.3%), and catechylpropanol C2 (2.2%) were all generated. Pyrocatechol C6 (1.1%) was also detected, being in line with the observation of hydrogenolysis of *vanilla* coats C-lignin[25,30]. In addition to catechols, depolymerized products from lipids and proteins, such as glycol and some amino acid derivatives could also be observed in small amounts. In 2D HSQC spectra of the soluble fraction, the signals corresponding to benzodioxane structures were no longer present[20,21], suggesting that the C-lignin had been depolymerized after catalytic reaction[25,30]. A family of correlation signals resonated at $\delta_C/\delta_H = 130.8/6.22$, 123.1/5.90, and 17.9/1.66 ppm were emerged, which were ascribed to the C=C bond and the methyl group in C4 propenyl group, respectively (Fig. 3f). The predominance of propenyl species in 2D NMR spectra is consistent with the observation in GC spectrum.

To evaluate the reusability of Ru/ZnO/C, the spent catalyst was subjected to simple washing with MeOH and was used directly in the following cycle under optimized conditions (Fig. 3d; Supplementary Table 6 and Supplementary Fig. 12). After six cycles, there was a slight degree of deactivation (monomers yield from 66 to 56%) but stable propenylcatechol selectivity (from 77% to 74.5%). Kinetic studies of C-lignin hydrogenolysis by using the fresh and spent catalyst were conducted at 180 °C under H₂, which resulted in comparable kinetic data (Supplementary Fig. 13). These results implied the stability of Ru/ZnO/C catalyst during hydrogenolysis. In the liquid samples, no leaching of Ru and Zn elements from the catalyst was detected by ICP-AES. In the spent Ru/ZnO/C catalyst after 1st run, Ru content was determined as 0.18 wt%, being slight lower than fresh Ru/ZnO/C (0.2 wt%) (Supplementary Table 1). The spent catalyst showed no obvious changes in their XRD and XPS spectra (Supplementary Figs. 1 and 2). The HAADF-STEM spectra displayed that Ru version in the spent Ru/ZnO/C was still dispersed as cluster and single atom without obvious aggregation (Supplementary Fig. 3).

Performing the hydrogenolysis reaction in the absence of catalyst or with ZnO/C obtained by the pyrolysis Zn-BTC led to poor yields of monomeric catechols (Fig. 3a). C7 with an allyl ether end-chain, probably derived from the etherification of caffeyl alcohol C1 with methanol, was observed in the catalyst-free condition (4.3%). Catechol C8, was also detected, which can

be suppressed in the presence of Ru/ZnO/C catalyst. 2D HSQC spectra showed that benzodioxane linkages still mostly remained in the absence of catalyst (Supplementary Fig. 14), implying a decisive role in the C−O bond cleavage of Ru specie, albeit in its low loading in Ru/ZnO/C. Hydrogenolysis of current C-lignin by commercial Ru/C (where Ru content is ca. 5 wt%) was also carried out (Fig. 3a)[25]. Catechol monomers were generated in 68% with 30 wt% Ru/C and 92% yields with 50 wt% Ru/C (Supplementary Table 4), respectively, from which similar TONs (11.4 and 9) of Ru/C were calculated. From the viewpoint of activity of Ru atoms, the use of Ru/ZnO/C (where TON is 278), is far higher than those from Ru/C-catalyzed hydrogenolysis of current castor C-lignin, as well as catalytic hydrogenolysis of vanilla C-lignin by Ru/C, Pt/C, and Pd/C (TONs = ca. 4)[25]. The intrinsic nature of atomic dispersion of Ru in Ru/ZnO/C should be responsible for the high activity, as ZnO/C was inactive for hydrogenolysis of C-lignin. In terms of product distribution, catechols having a saturated end-chain, including propylcatechol C3 and catechylpropanol C2 were the two major products by Ru/C, being consistent with the previous report[25]. Compared with Ru/C, less retention of γ-OH over Ru/ZnO/C was probably because ZnO species facilitates the hydrogenolysis of primary alcohol[18] and/or less hydrogenation affinity of Ru/ZnO/C gives a chance for preferential hydrogenolysis.

To further understand the performance of Ru/ZnO/C, the hydrogenolysis reactions were performed under various conditions. The choice of other solvents, such as EtOH, iPrOH, dioxane, and tetrahydrofuran (THF) gave decreased yields of catechyl monomers, whereas the selectivity toward propenylcatechol remained high (Fig. 3a) The total monomers yields approximately followed a linear relationship with the solvent polarity ($E_T(30)$) (Supplementary Fig. 15)[49], which is similar to the observation of hydrogenolysis of β-O-4 type lignin[50]. In the cases of less-polar solvents (iPrOH, dioxane and THF), caffeyl alcohol C1, the starting monolignol for C-lignin biosynthesis, was observed (Supplementary Table 7). Raising the reaction temperature led to the continuous production of monomeric catechols, finally reaching 82% yield at 240 °C (Fig. 4a). Meanwhile, the selectivity switched from propenylcatechol C4 (200 °C, 77%) to propylcatechol C3 (240 °C, 72%) because the hydrogenation of C=C bond is enhanced under harsher conditions. The yield and selectivity to C3 by Ru/ZnO/C under such a condition were comparable to those from hydrogenolysis of C-lignin by 50 wt%-dosage of Ru/C (90% yield, 74% selectivity to C3), while the calculated TON was superior (345 vs 9). Similar trends were also observed in the case of prolonging the reaction time or increasing the catalyst dosage (Fig. 4b, c). These results implied that Ru/ZnO/C does enable to the hydrogenation of C=C bond, but it is posterior to the cleavage

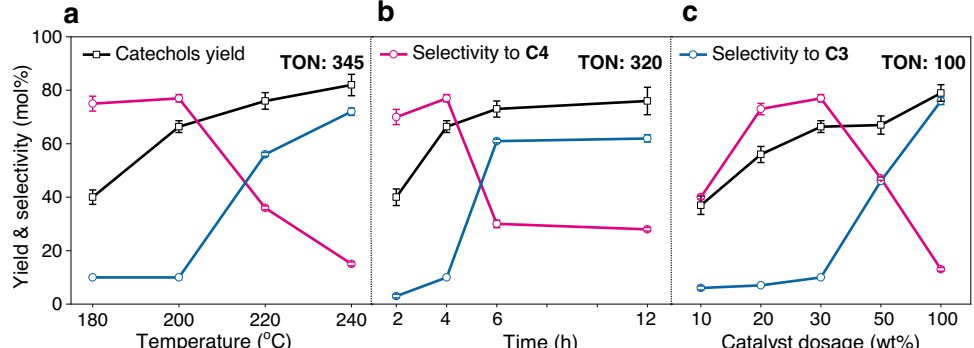

**Fig. 4 Parameters effects. a** Reaction temperature effect, conditions: C-lignin (50 mg), Ru/ZnO/C (30 wt%), MeOH (10 mL), 3 MPa H₂, 4 h. **b** Reaction time effect, at 200 °C, 30 wt% Ru/ZnO/C. **c** Catalyst dosage effect, at 200 °C, 4 h. (Error bars = standard deviation; *n* = 2; for 200 °C, 4 h and 30 wt% of Ru/ZnO/C, *n* = 3; If no error bars are visible, they are smaller than the corresponding symbol size).

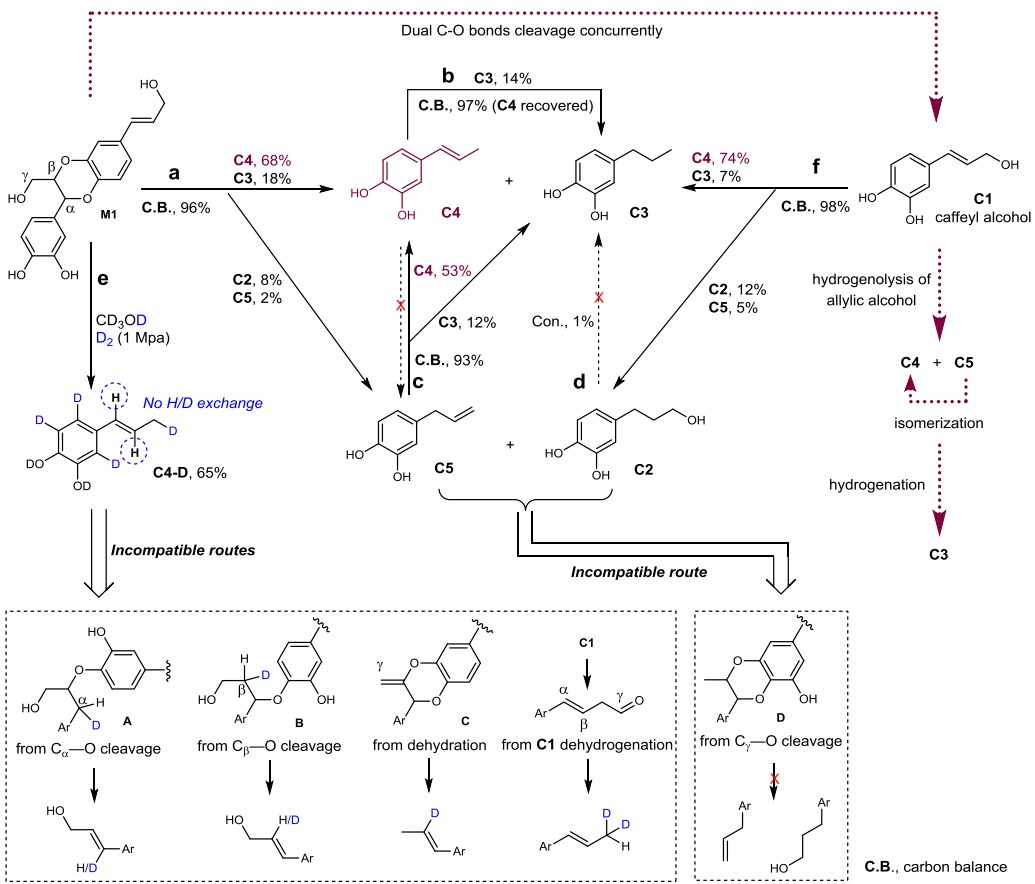

**Fig. 5 Ru/ZnO/C-catalyzed hydrogenolysis of C-lignin model and catechols.** Reaction conditions: substrate (50 mg), Ru/ZnO/C (10 mg, 20 wt%), MeOH (10 mL), 1 MPa $H_2$, 4 h, unless otherwise noted. **a** M1 as substrate (reaction a). **b** C4 as substrate (reaction b). **c** C5 as substrate (reaction c). **d** C2 as substrate (reaction d). **e** Reaction in $D_2$ and $CD_3OD$ with M1 as substrate (reaction e), and other catechols were also generated. **f** C1 as substrate (reaction f).

of dual C−O bonds in benzodioxane units under a $H_2$ atmosphere. From a chemodivergent standpoint, Ru/ZnO/C allowed for producing catechol with a propenyl or propyl end-chain selectively through varying reaction conditions.

**Mechanistic insights**. We were intrigued about the mechanism of C-lignin hydrogenolysis, as it has scarcely been discussed in the literature[25,30]. Mechanistic insights were obtained from the reactivity studies of a C-lignin model compound M1 and corresponding catechols (Fig. 5). Upon the treatment of M1 with Ru/ZnO/C at 200 °C and $H_2$, propenylcatechol C4 was generated as the major product (68%), along with the observation of catechylpropanol C2 (8%), propylcatechol C3 (18%), and allylcatechol C5 (2%) (reaction a). The products distribution was kin to those from catalytic hydrogenolysis of C-lignin biopolymer. The treatment of C5 led to the formation of C4 primarily (53%) through isomerization and propylcatechol subsidiarily (12%) through hydrogenation (reaction c). No isomerization from C4 to C5 was observed (reaction b). The low conversion of C4 or C5 to C3 (12 and 14% yields) confirmed again that the hydrogenation of either internal or terminal C=C bond is a non-favorable route under such a condition. Independent reaction of C2 with Ru/ZnO/C failed to result in any detectable products, excluding the possibility of that C2 acts as an intermediate to C3 (reaction d). Non-negligible amounts of catechylpropanol C2 and allylcatechol C5 resulted from biopolymer and dimeric model M1 reactions suggested that a possible intermediate D derived from the preferential cleavage of $C_γ$−O is infeasible. We further carried out

the hydrogenolysis of M1 in a deuterated condition, that is, in $CD_3OD$ and $D_2$ atmosphere (reaction e). The [1]H NMR spectrum analysis of isolated C4-D suggested that the protons at α, β, and γ positions were kept intact, despite complete H/D exchange reaction has occurred on aromatic ring (Supplementary Fig. 21). The intactness of γ protons illustrated that the cleavage of $C_γ$−OH bond follows a hydrogenolysis route rather than a dehydrogenation route (to form an aldehyde)[51]. The well-preserved α-H and β-H implied the linkage protons do not take part in the decomposition of benzodioxane units directly. Thereby, the possible reaction pathways, such as the hydrogenolysis of $C_α$−O (leading to A) or $C_β$−O bond (leading to B) followed by an elimination reaction, can be completely excluded because half proton at α or β position would be replaced with deuterium[52]. Similarly, a possible intermediate C, derived from an initial dehydration reaction, is not associated with reaction pathway. In view of the integrality of $C_α$−H and $C_β$−H, together with the generation of a C=C bond in C4, we postulated that the $C_α$−O and $C_β$−O bonds, which have similar enthalpies[29] are cleaved at the same time. In this context, caffeyl alcohol C1, which could be observed in C-lignin hydrogenolysis in $^iPrOH$, dioxane, and THF, emerged as a possible intermediate. To test this, we treated C1 with Ru/ZnO/C (reaction f). This reaction occurred in a fashion akin to that of hydrogenolysis of C-lignin biopolymer and model compound M1, that is, yielding C4 in 74%, C2 in 12%, C3 in 7%, and C5 in 5%, respectively. The consistency in products distribution approved the rationality of caffeyl alcohol as an intermediate. No detection of C1 in MeOH forecasted that the cleavage of $C_γ$−OH bond of C1 should be

faster than the cleavage of C−O bonds of benzodioxane unit, and this was substantiated by the parallel treatment of **C1** and **M1** with Ru/ZnO/C at 130 °C, where **C1** was converted into **C4** (76%) and **M1** remained unchanged (Supplementary Fig. 22).

Based on above results, a plausible pathway for Ru/ZnO/C-catalyzed hydrogenolysis of C-lignin was rationalized (red line). Initially synchronous cleavage of $C_\alpha$−O and $C_\beta$−O bonds of benzodioxane structure probably through a concerted process results in caffeyl alcohol **C1**. Hydrogenolysis of allylic alcohol of **C1** gives propenylcatechol **C4** and allylcatechol **C5**, which should be faster than the cleavage of benzodioxane structure as well as the hydrogenation of the C=C bond[53,54]. Given that the instability of caffeyl alcohol, hydrogenolysis of $C_\gamma$ − OH may act as a stabilization mechanism against its fast repolymerization[55]. The isomerization of **C5** leading to **C4** is irreversible reaction, and both of them could be hydrogenated into **C3** slowly. Overall, the present Ru/ZnO/C catalyst featured the high C−O bond hydrogenolysis affinity rather than C=C bonds hydrogenation, and this feature distinguished from other catalysts makes for the generation of propenylcatechol efficiently and selectively.

## Discussion

In summary, we prepared an atomic dispersion ruthenium catalyst by the in situ anchoring Ru in MOFs and pyrolysis strategy. The characterization by HAADF-STEM and XAFS showed the dispersion of Ru species in either clusters or single atoms manner. This catalyst was capable to cleave all C−O bonds in benzodioxane structures efficiently and circumvent the reduction of C=C bonds selectively, thus depolymerizing C-lignin biopolymer into propenylcatechol preferentially. The activity of current Ru catalyst is far higher than that of commercial Ru/C in term of producing catechols, which, together with the selectivity to propenylcatechol can be maintained at least six cycles. Mechanistic studies by model compounds excluded the pathways containing dehydrogenation and/or dehydration reactions, and deduced a possible route with caffeyl alcohol as an intermediate. Successful demonstration of selective hydrogenolysis of C-lignin into propenylcatechol validates the utility of atomically dispersed metal catalysts in the development of abundant renewable biopolymer conversion technologies.

## Methods

**Chemicals**. All commercially available chemicals were used as received, unless otherwise stated. Castor seed coats were obtained from Shandong, China. Cellulase (NS22086, 1000 BHU(2) g$^{-1}$) and xylanase (NS22083, 2500 FXU-S g$^{-1}$) were provided from Novozymes. The model compound and authentic samples were synthesized independently.

**Preparation of Ru/ZnO/C catalyst**. 1,3,5-Benzenetricarboxylic acid (1.26 g, 6 mmol), Zn(OAc)$_2$ (0.91 g, 5 mmol) and RuCl$_3$·3H$_2$O (5 mg, 0.02 mmol) were mixed in N,N-dimethylformamide (DMF, 50 mL) at room temperature. After stirring at 140 °C for 12 h, the precipitate was collected by centrifugation and washed with fresh DMF, which then underwent drying under vacuum at 120 °C for 10 h. The dried powder was transferred into a ceramic boat and calcined at 750 °C for 2 h in a stream of N$_2$, thus producing the Ru/ZnO/C catalyst. The as-prepared catalyst was stored in a glove-box for further use.

**Catalyst characterizations**. Ru/ZnO/C samples were prepared for ICP-AES analysis by dissolution in aqua regia solution, which were then measured on a Thermo IRIS Intrepid II spectrometer. HAADF-STEM images and elemental mapping were obtained on a JEOL JEM-ARM 200 F TEM/STEM system at 200 kV. XRD data were obtained on a Rigaku Miniflex-600 under operating at 40 kV voltage and 15 mA current with Cu Kα radiation ($\lambda$ = 0.15406 nm). XPS was measured on a ThermoFischer ESCALAB 250Xi X-ray photoelectron spectrometer with Al Kα radiation, in which carbon was used as an internal standard (C1s = 284.6 eV). XAFS spectra at the Ru (ruthenium) K-edge were collected at BL14W1 station in Shanghai Synchrotron Radiation Facility (SSRF), the detailed analysis process was shown at Supplementary information. Elemental analysis was performed in a vario MACRO cube (Elementar Analysensysteme GmbH, Germany) with CHN mode.

**Isolation and characterization of C-lignin**. The castor seed coats (endocarps) (10 g, 20–40 mesh) were extracted with acetone (8 h) and water (5 h) to remove some extractives (ca. 15%) using a Soxhlet instrument, which was then treated with cellulase (5 mL) and xylanase (1 mL) in 200 mL citrate buffer (50 mM, pH = 4.8) in a shaking bath at 50 °C for 48 h twice. The collected solid residue was suspended in a mixture of dioxane and HCl aqueous solution (85:15 v/v, 100 mL) and refluxed under nitrogen for 4 h. After filtration, the soluble fraction was neutralized with NaHCO$_3$, followed by evaporation to afford a thick solution. Upon the treatment of HCl aqueous solution (pH = 2.0), a precipitate was formed, which was allowed to equilibrate at 4 °C overnight. The C-lignin sample was collected by centrifugation, washing with HCl aqueous solution (pH = 2.0), and freeze–drying (1.2 g).

2D HSQC NMR spectra were acquired on a Bruker Avance 400 MHz spectrometer by using a C-lignin sample (50 mg) dissolved in dimethyl sulfoxide (DMSO)-$d_6$ (0.5 mL). The cross signals were assigned based on NMR data of the model compounds and the available literature data. Quantitative $^{13}$C NMR analysis used a C-lignin sample (100 mg) dissolved in DMSO-$d_6$ (1 mL) with 1,3,5-trioxane as an internal standard, and the relaxation time of NMR spectrometer was set as 12 s under deNOE mode. The average molecular weight was performed on Shimadzu LC-20AD equipped with a PL-gel 10 μm Mixed-B 7.5 mm I.D. column and UV detection detector (254 nm) at 50 °C, using THF as the solvent, which was calibrated with polystyrene standards (Polymer Laboratories Ltd.).

Thioacidolysis analysis was performed according to previous reports[20,21]. In brief, 30 mg samples (extractive-free castor seed coats or isolated lignin samples) were treated with 5 mL of 0.2 M BF$_3$-etherate in an 8.75:1 dioxane/ethanethiol mixture at 100 °C for 4 h. The reaction mixture was extracted by dichloromethane. After the removal of all volatiles of the organic phase under vacuum, the resulted residue was dissolved in anhydrous THF, which was treated with BSTFA at 65 °C for 1 h under N$_2$ before GC/MS analysis.

**Hydrogenolysis of C-lignin**. In a typical reaction, C-lignin sample from endocarp of castor seed coats (50 mg), Ru/ZnO/C catalyst (15 mg, 30 wt%) and methanol (10 mL) were charged in a 50 mL Parr autoclave. The reactor was sealed, purged with nitrogen, and then was pressured to H$_2$ (3 MPa) at room temperature. The reaction was carried out at different temperatures for a certain time with a magnetic stirring. After reaction, the autoclave was cooled and depressurized carefully. The insoluble fraction was removed by filtration, and lignin oily product was obtained from the soluble fraction after the removal of all volatiles under vacuum conditions.

An external standard (tetradecane) was added to the lignin oily solution in anhydrous THF, which was treated with BSTFA at 65 °C for 1 h under N$_2$. The resulted mixture was subjected to Shimadzu GCMS-QP2010SE equipped with an HP-5 MS column and Shimadu GC-2010 equipped with an HP-5 column for analysis. The identification and quantification of catechol monomers in the oily product were assessed by comparison with authentic samples from independent synthesis.

## Data availability

The source data underlying Figs. 2d–e, 3a–d, and 4 and Supplementary Figs. 1, 2, 4, 8, 10, 12, 13, 15, 16, 18–20 is provided as a Source Data file. The data that support the findings of this study are available from the corresponding author upon reasonable request. Source data are provided with this paper.

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

## Acknowledgements
This work was supported by the National Natural Science Foundation of China (31971607, 21776020), the Natural Science Foundation of Liaoning Province (no. 2019-MS-019), and the Science Foundation of Dalian Polytechnic University (no. 61010201). We thank Dr. Wenxing Chen (Beijing Institute of Technology) for the aid of XAFS analysis.

## Author contributions
G.S. and S.W. conceived the project. S.W. and K.Z. designed the synthesis of the catalysts. S.W., H.L., and L.-P.X. performed the characterizations and catalytic reactions. G.S. and S.W. wrote the manuscript. All authors commented on the manuscript.

## Competing interests
The authors declare no competing interests.
