## [Peer Review File · Nature Communications]

REVIEWER COMMENTS

Reviewer #1 (Remarks to the Author):

The manuscript describes the valorization of a lignin composed entirely of caffeyl alcohol units, termed C-lignin, using a heterogeneous Ru-based heterogeneous catalyst under reductive conditions, yielding propenylcatechol with high selectivity. The authors perform a mechanistic investigation of their reaction conditions using dimeric and monomeric relevant model compounds, revealing caffeyl alcohol as an important intermediate and the synchronous cleavage of the benzodioxane C α -O and C β -O linkages. The valorization of isolated C-lignin to catechols in high yields has been achieved, thus the novelty lies in the increased efficiency of the developed catalyst and in the mechanistic work. Although this article might be better suited for a more specialized journal, such as Nature Catalysis or a sustainability journal, the work is pertinent to Nature Communication's mission and will be valuable for the lignin valorization community. The authors perform a rational research plan, the manuscript has a logical flow, and the analytical measurements presented are solid, but some additional analytical information is necessary for the manuscript to have reproducible data, for example on the lignin isolation procedure. Furthermore, several points should be adjusted before consideration for publication:

- 1) Several typos are present in the manuscript, including in figures, which at times make understanding difficult (such as in the mechanism section, do the authors mean deoxygenation?)
- 2) The authors developed a novel catalyst based on Ru in order to increase efficiency of benzodioxane reductive cleavage while retaining aromatic units and unsaturation in product side chains. Could the authors provide the rationale on the choice of Ru as the active metal? Reports of Guaiacyl/syringyl lignin depolymerization has demonstrated that cheaper transition metals can also be efficient yet more selective, due to their lack of reactivity toward arene hydrogenation.
- 3) Could the authors provide the NMR solvents used and conditions for each reported spectra?
- 4) The authors use a sequence of known procedures for the isolation of their C-lignin sample from castor seed coats – enzymatic and mild acidolysis. Could the authors specify the procedures in the SI? The referenced work does not provide it.
Reporting the full procedures is important because it affects the following points:
 - a) Castor seed shells are known to also contain guaiacyl and syringyl -type lignin. Did the authors assess how much of the G/S lignin is present in their C-lignin sample? Their lignin isolation procedure is based on the dissolution of the carbohydrates, leaving lignin behind as the solid. It is thus unlikely that a selective C-lignin separation occurs. This should at least be discussed in the manuscript, or G/S lignins should be quantitated for example by thioacidolysis.
 - b) Discounting the presence of G/S lignins is crucial because those lignins may express a signal in the ¹³C NMR from B-O-4 linkages around the same chemical shift as the chosen benzodioxane signal that the authors use for quantitating C-lignin in their sample. The potential presence of G/S lignin would change (increase) their reported molar yields. Perhaps the authors can report yields in wt% if G/S lignins are

present

5) Could the authors report what else is present in the product oil (65 mol% 4-propylcatechol \approx 35-40 wt% of the oil) – oligomers, other monomers, etc. ?

Reviewer #2 (Remarks to the Author):

General

In this manuscript, the author reports an atomically dispersed Ru catalyst for the hydrogenolysis of C-lignin to produce catechols via the concurrent cleavage of C-O bonds in benzodioxane rings. The Ru species are dispersed as clusters or single atoms verified by HAADF-STEM and XAFS, and it appears to be the first time that atomically dispersed metal catalyst are being applied to macromolecule transformation. The author used HSQC, GPC and GC to characterize and quantify the products, showing that Ru/ZnO/C can effectively avoid the reduction of C=C bonds compared with commercial Ru/C catalyst, giving propenylcatechol as the primary product with a high TON. They also studied the product distribution of hydrogenolysis of C-lignin catalysed by Ru/ZnO/C under different reaction conditions (changing solvent, reaction temperature, reaction time and catalyst dosage). Interestingly, the reaction mechanism has also been investigated by using synthetic model compounds and deuterated reagents. They deduced a possible route with caffeyl alcohol as an intermediate and excluded the pathways containing dehydrogenation and elimination reactions. I recommend the publication of this work after addressing the comments below.

Minor comments

1. Line 164

The author said that the calculation of TON is based on dispersion of Ru. "Dispersion" is a concept which describe the fraction of atoms exposed to the surface. It seems that they used the ICP results for the calculation of Ru/ZnO/C, which is reasonable because of its atomically dispersed nature. However, for the TON of Pt/C, Pd/C and Ru/C in the cited paper, it seems that the calculation is based on the total number of moles of noble metal, which is not accurate given that these are not atomically dispersed. If both determinations of TON are based on the total moles of metals, perhaps use this instead of "dispersion" as it is not accurate in the latter case.

2. Line 286

The author explained that the removal of caffeyl alcohol at the γ -position is probably assisted by the coordination of Ru with the C=C bond. Is there any evidence or further explanation to clarify why such coordination structure would form? Otherwise please cite related references.

3. Line 184

"The spent catalyst showed no obvious changes in their XRD and XPS spectra." Please indicate the figure

number.

4. Figure 3

- a. In compound A (Benzodioxane), It would look neater if the double bond of the aromatic ring was displayed internally to the aromatic ring.
- b. In Fig. c, there is a spelling mistake in “retention time”.

Reviewer #3 (Remarks to the Author):

This is an interesting study on the catalytic hydrogenolysis of C-lignin. The authors prepare Ru/ZnO/C catalyst by pyrolysis of ZnO MOF in presence of RuCl₃. The characterization data seem ok but I could not find the exact data of ICP-AES and therefore complete elemental analysis is missing. How much Cl of the RuCl₃ precursor remains in the catalyst sample?

The catalyst shows a high selectivity for the formation of propenylcatechol (C₄), which is indeed remarkable. Still the authors report GC yields only, and this is of a mixture of products. To claim real yields isolation of the isolated pure product is required. Also, proper mass balances based on the weight of the samples added to the reactor must be provided. The total amount of oily product does not match the total sum of the identified products in many cases. What are the missing components, are they non-volatile products not visible on GC? Does the solid residue plus oily product add-up to 100% of the lignin sample used?

When the reaction is followed in time the amount of oily product decreases after 4 hours whereas the total yield of identified products increases. How can that be? This should be explained.

The mechanistic study using deuterium labeling provides interesting information. Nevertheless, the conclusion of the authors that degradation pathway goes via the caffeyl alcohol C₁ is in contradiction with absence of this product in all their experiments with their catalyst, even at low T (Table S6) or short reaction time (Table S7). They did try C₁ as substrate and see similar product distribution but the important kinetics are missing. If they are correct that reaction proceeds via C₁ while it is never observed as product, conversion of C₁ should be orders of magnitude faster than degradation of C-lignin or model compound. This information should be provided. Also, they could analyse the intermediate reaction mixture with 2-D NMR to identify all intermediates present.

Furthermore, they report formation of C₆, C₇ and C₈ but ignore them in their discussion. Especially C₈ is formed initially but disappears in time (Table S7). Why is that? Is this an intermediate in their catalytic process?

The authors suggest that C₃ is formed via hydrogenation of C₄. Why does this not occur in the first part of the reaction? The report 28% C₄ after 2 h but only 1% C₃ (Table S7). After 4 h they have 51% C₄ and 6.6 C₃. But then after 6 h they gain more than 38% C₃, while the lose only 31% C₄. So gain of C₃ cannot be explained by loss of C₄, at least not completely.

They have performed the recycling studies at reaction times of 4 hours where they have already reached their maximum yield and therefore this is not proof of catalyst stability. Again, they should provide proper kinetics or at least do this at lower conversion. They should provide data about Ru and Zn leaching in the liquid phase and recheck the elemental composition of their spent catalysts.

In conclusion, I believe this work might be publishable but only after major revisions including additional experimental data.

The point-by-point response to the reviewers' comments

Reviewer #1 (Remarks to the Author):

The manuscript describes the valorization of a lignin composed entirely of caffeyl alcohol units, termed C-lignin, using a heterogeneous Ru-based heterogeneous catalyst under reductive conditions, yielding propenylcatechol with high selectivity. The authors perform a mechanistic investigation of their reaction conditions using dimeric and monomeric relevant model compounds, revealing caffeyl alcohol as an important intermediate and the synchronous cleavage of the benzodioxane C α -O and C β -O linkages. The valorization of isolated C-lignin to catechols in high yields has been achieved, thus the novelty lies in the increased efficiency of the developed catalyst and in the mechanistic work. Although this article might be better suited for a more specialized journal, such as Nature Catalysis or a sustainability journal, the work is pertinent to Nature Communication's mission and will be valuable for the lignin valorization community. The authors perform a rational research plan, the manuscript has a logical flow, and the analytical measurements presented are solid, but some additional analytical information is necessary for the manuscript to have reproducible data, for example on the lignin isolation procedure. Furthermore, several points should be adjusted before consideration for publication:

1) Several typos are present in the manuscript, including in figures, which at times make understanding difficult (such as in the mechanism section, do the authors mean deoxygenation?)

Reply: Thanks very much for these suggestions. After careful checking, some typos in manuscript and figures have been revised.

In the mechanism section, a reported pathway for C γ -OH cleavage in **C1** undergoes a dehydrogenation step (oxidation) to give an aldehyde specie. We have modified the description.

2) The authors developed a novel catalyst based on Ru in order to increase efficiency of benzodioxane reductive cleavage while retaining aromatic units and unsaturation in product side chains. Could the authors provide the rationale on the choice of Ru as the active metal? Reports of Guaiacyl/syringyl lignin depolymerization has demonstrated that cheaper transition metals can also be efficient yet more selective, due to their lack of reactivity toward arene hydrogenation.

Reply: As hydrogenolysis of C–O bonds is metal dependent, depolymerization of benzodioxane structures in C-lignin may likely be accomplished by the choice of a metal center having strong activity to hydrogenolysis C–O bonds, such as Ru.

Actually, we have scanned some heterogeneous metal catalysts. For example, supported MoO₂ catalyst which can efficiently depolymerize guaiacyl/syringyl (β -O-4) lignin into unsaturated phenolic monomers (ref. 17), did not show any activity for C-lignin hydrogenolysis. We also used palladium (Pd) metal with a lower activity toward C-O bond hydrogenolysis as active metal, which led to negative results in C-lignin hydrogenolysis.

The description on the rationale for the choice of Ru as the active metal has been added in manuscript. We thank this reviewer for this important suggestion.

3) Could the authors provide the NMR solvents used and conditions for each reported spectra?

Reply: Deuterated solvents for NMR spectra have been added in manuscript and SI. Thanks for this prompt.

4) The authors use a sequence of known procedures for the isolation of their C-lignin sample from castor seed coats – enzymatic and mild acidolysis. Could the authors specify the procedures in the SI? The referenced work does not provide it.

Reply: The enzymatic and mild acidolysis procedure for C-lignin extraction has been described in the manuscript (Methods).

Reporting the full procedures is important because it affects the following points:

a) Castor seed shells are known to also contain guaiacyl and syringyl-type lignin. Did the authors assess how much of the G/S lignin is present in their C-lignin sample? Their lignin isolation procedure is based on the dissolution of the carbohydrates, leaving lignin behind as the solid. It is thus unlikely that a selective C-lignin separation occurs. This should at least be discussed in the manuscript, or G/S lignins should be quantitated for example by thioacidolysis.

Reply: In this manuscript, we used endocarp of castor seed coats (as shown below), where we did not find the guaiacyl and syringyl-type lignin (β -O-4-type lignin) based on the analysis of 2D NMR and ^{13}C NMR spectra, as well as the analysis of depolymerized products from hydrogenolysis reactions. Of note, castor seed epicarps contain guaiacyl and syringyl-type lignin, as well as hydroxycinnamic species, where no C-lignin could be detected.

Current EMAL method for extraction C-lignin is a mild procedure, by which β -O-4 structures can be maintained well (see ref. 17); thereby, the possibility of destroying β -O-4 lignin during its extraction procedure could be excluded. Previous report has described the coexistence of C-lignin and β -O-4 lignin in some other seed coats (endocarp), which could be identified by 2D NMR spectra (ref. 21).

We thank this suggestion, and some discussions have been added in manuscript and SI.

b) Discounting the presence of G/S lignins is crucial because those lignins may express a signal in the ^{13}C NMR from $\beta\text{-O-4}$ linkages around the same chemical shift as the chosen benzodioxane signal that the authors use for quantitating C-lignin in their sample. The potential presence of G/S lignin would change (increase) their reported molar yields. Perhaps the authors can report yields in wt% if G/S lignins are present.

Reply: In ^{13}C NMR spectra of C-lignin from castor seed coats, we could not observe the signals from $\beta\text{-O-4}$ structures. The phenolic products derived G/S lignins were not observed during the hydrogenolysis. The detailed explanations were showed as above.

5) Could the authors report what else is present in the product oil (65 mol% 4-propylcatechol \approx 35-40 wt% of the oil) – oligomers, other monomers, etc.?

Reply: The reviewer's point is right, because current isolated C-lignin still contained some unknown species. The molar concentration of caffeoyl alcohol units was measured as 2.49 mmol g^{-1} based on the analysis of quantitative ^{13}C NMR spectroscopy, which corresponded to 41 wt% of current isolated C-lignin sample. Lipids and proteins might be other species in isolated C-lignin from castor seed coats, as seeds serve a different role in the plant than traditional lignocellulosic material. We found glycol and some amino acids in product oil after hydrogenolysis, which may derive from lipids and proteins.

The mass balance, that is the solid residue plus oily product, were calculated, which ranged from 85% to 97% based on the weight of C-lignin samples. These data have been added in the SI.

In GPC spectra, some oligomers and dimers could be observed in the oily product, which showed less abundance than those from Ru/C catalyst.

Reviewer #2 (Remarks to the Author):

General

In this manuscript, the author reports an atomically dispersed Ru catalyst for the hydrogenolysis of C-lignin to produce catechols via the concurrent cleavage of C-O bonds in benzodioxane rings. The Ru species are dispersed as clusters or single atoms verified by HAADF-STEM and XAFS, and it appears to be the first time that atomically dispersed metal catalyst are being applied to macromolecule transformation. The author used HSQC, GPC and GC to characterize and quantify the products, showing that Ru/ZnO/C can effectively avoid the reduction of C=C bonds compared with commercial Ru/C catalyst, giving propenylcatechol as the primary product with a high TON. They also studied the product distribution of hydrogenolysis of C-lignin catalysed by Ru/ZnO/C under different reaction conditions (changing solvent, reaction temperature, reaction time and catalyst dosage). Interestingly, the reaction mechanism has also been investigated by using synthetic model compounds and deuterated reagents. They deduced a possible route with caffeyl alcohol as an intermediate and excluded the pathways containing dehydrogenation and elimination reactions. I recommend the publication of this work after addressing the comments below.

Minor comments

1. Line 164

The author said that the calculation of TON is based on dispersion of Ru. "Dispersion" is a concept which describe the fraction of atoms exposed to the surface. It seems that they used the ICP results for the calculation of Ru/ZnO/C, which is reasonable because of its atomically dispersed nature. However, for the TON of Pt/C, Pd/C and Ru/C in the cited paper, it seems that the calculation is based on the total number of moles of noble metal, which is not accurate given that these are not atomically dispersed. If both determinations of TON are based on the total moles of metals, perhaps use this instead of "dispersion" as it is not accurate in the latter case.

Reply: The reported TONs from Ru/ZnO/C in this manuscript was calculated base the total number of moles of Ru metal (Ru content in catalyst: 0.2 wt%). The description of "based on the dispersion of Ru on the catalyst" is not right, and it has been revised. We thank for pointing out this mistake.

2. Line 286

The author explained that the removal of caffeyl alcohol at the γ -position is probably assisted by the coordination of Ru with the C=C bond. Is there any evidence or further explanation to clarify why such coordination structure would form? Otherwise please cite related references.

Reply: In homogeneous catalysis, hydrogenolysis of allylic alcohols (C γ -OH cleavage) assisted by the coordination of Co with C=C bond has been reported (*J. Chem. Soc. Chem. Commun.* **1978**, 63). In view of the non-reactivity of **C2** bearing a saturated alcohol, we supposed that

hydrogenolysis of unsaturated alcohol in **C1** may proceed in the coordination of Ru with C=C bond step.

Given that we do not have strong evidence to support this propose now, this description and the corresponding transition state in Fig. 5 were removed from the manuscript. Two references on hydrogenolysis of allylic alcohols were added.

Ref: 53. *J. Chem. Soc., Chem. Commun.*, 63-65 (1978);

54. *Chem. Eur. J.* **23**, 18025-18032 (2017).

Thanks for this question.

3. Line 184

“The spent catalyst showed no obvious changes in their XRD and XPS spectra.” Please indicate the figure number.

Reply: The Figure numbers have been added as Supplementary Figs. 1 and 2 in the manuscript.

4. Figure 3

a. In compound A (Benzodioxane), It would look neater if the double bond of the aromatic ring was displayed internally to the aromatic ring.

b. In Fig. c, there is a spelling mistake in “retention time”.

Reply: Thanks for pointing out these errors. We have revised these in the manuscript.

Reviewer #3 (Remarks to the Author):

This is an interesting study on the catalytic hydrogenolysis of C-lignin. The authors prepare Ru/ZnO/C catalyst by pyrolysis of ZnO MOF in presence of RuCl₃. The characterization data seem ok but I could not find the exact data of ICP-AES and therefore complete elemental analysis is missing. How much Cl of the RuCl₃ precursor remains in the catalyst sample?

Reply: ICP-AES data have been updated in the SI. For the fresh Ru/ZnO/C, Ru and Zn contents were measured as 0.2 wt% and 51wt%; for the spent Ru/ZnO/C (after 1st use), Ru and Zn were measured as 0.18 wt% and 48 wt%. Chlorine was not found in fresh Ru/ZnO/C by XPS (solid) and ion chromatogram (solution in nitric acid), probably it has been released as HCl during thermal treatment.

The catalyst shows a high selectivity for the formation of propenylcatechol (C4), which is indeed remarkable. Still the authors report GC yields only, and this is of a mixture of products. To claim real yields isolation of the isolated pure product is required. Also, proper mass balances based on the weight of the samples added to the reactor must be provided. The total amount of oily product does not match the total sum of the identified products in many cases. What are the missing components, are they non-volatile products not visible on GC? Does the solid residue plus oily product add-up to 100% of the lignin sample used?

Reply: At 200° C, 3 MPa of H₂ in MeOH for 4 h, when 100 mg C-lignin was used, propenylcatechol **C4** could be isolated with 18 mg (18% yield based on the weight of C-lignin sample; 0.12 mmol, 48 mol% yield based on the caffeyl alcohol units) by silica gel chromatographic column. This result has been added in the manuscript, and detailed procedure has been added in the SI.

After hydrogenolysis, solid recovery contained catalyst and unreacted material (most may be lipids, proteins and other unidentified species). The mass balance, that is the solid unreacted material plus oily product, were calculated, which ranged from 85% to 97% based on the weight of C-lignin samples. These data have been added in the SI.

Quantitative ¹³C NMR spectroscopy gave the molar concentration of caffeyl alcohol units as 2.49 mmol g⁻¹, which corresponded to 41 wt% of current isolated C-lignin sample. Of note, current 2.49 mmol g⁻¹ concentration is higher than that from C-lignin sample isolated from vanilla seed coats (1.1 mmol g⁻¹, ref. 25.). As seeds serve a unique role in the plant than traditional lignocellulosic material, lipids and proteins might be other species in current C-lignin sample. In product oil after hydrogenolysis, we can only find glycol and some amino acid derivatives in a small amount. Other products could not be detected and identified by GC and NMR (2D and ¹³C).

We really appreciate this suggestion, which make the manuscript clearer.

When the reaction is followed in time the amount of oily product decreases after 4 hours whereas the total yield of identified products increases. How can that be? This should be explained.

Reply: The phenomenon mentioned by the reviewer, that the decrease of oily product with the time (4 h, 70%; 6 h, 68%; 12 h, 60%) is right. We also noticed that the oily product yields decreased with the raising of reaction temperature (200 °C, 70%; 240 °C, 61%). Actually, mass balance also decreased under harsher conditions.

We suspected that further transformation on derivatives from lipids and proteins (such as decarboxylation of amino acid leading to the release CO₂ and/or deoxygenation leading to H₂O) led to the decrease of the oily product yields under harsher conditions. We carried out the elemental analysis of C-lignin sample and resulted oily product. We found the contents of carbon and nitrogen were both increased in the oily product, suggesting that some oxygen atoms have been emitted during hydrogenolysis (Supplementary Table 5).

The mechanistic study using deuterium labeling provides interesting information. Nevertheless, the conclusion of the authors that degradation pathway goes via the caffeyl alcohol C1 is in contradiction with absence of this product in all their experiments with their catalyst, even at low T (Table S6) or short reaction time (Table S7). They did try C1 as substrate and see similar product distribution but the important kinetics are missing. If they are correct that reaction proceeds via C1 while it is never observed as product, conversion of C1 should be orders of magnitude faster than degradation of C-lignin or model compound. This information should be provided. Also, they could analyse the intermediate reaction mixture with 2-D NMR to identify all intermediates present.

Reply: When ¹PrOH, dioxane or THF was used as a solvent, caffeyl alcohol **C1** could be observed with 3-4% yield in Supplementary Table 7.

We accept the reviewer's point that "conversion of **C1** should be orders of magnitude faster than degradation of C-lignin or model compound." To confirm this, **C1** and dimeric model **M1** were treated with Ru/ZnO/C (30 wt%) at 130 °C under H₂ (2 MPa, RT) for 3 h, respectively. Under such a condition, caffeyl alcohol **C1** underwent hydrogenolysis of allylic alcohol to give **C4** in 76% yield, while **M1** remained nearly intact (based on the analysis of ¹H NMR spectra of crude products). These results substantiated that the cleavage of C_V-OH in **C1** is faster than the cleavage of C-O bonds of benzodioxane structure.

We thank this reviewer for the insightful advice, and these results have been added in the manuscript and SI.

Note: Tables S6 and S7 have updated as Tables S8 and S9 in revised SI now.

Furthermore, they report formation of C6, C7 and C8 but ignore them in their discussion. Especially C8 is formed initially but disappears in time (Table S7). Why is that? Is this an intermediate in their catalytic process?

Reply: Pyrocatechol **C6** could be observed with 1-3% yield in most hydrogenolysis reactions of C-lignin biopolymer, while it was absent in dimeric model reactions. Thereby, we proposed that **C6** may derive from intrinsic pyrocatechol units in current C-lignin biopolymer.

Compound **C7** is an etherified product of caffeoyl alcohol with methanol, which could be detected under control reaction.

The mass value of **C8** was determined as 292 g mol⁻¹ after silylation, being 2 g mol⁻¹ less than **C4** (294 g mol⁻¹ after silylation). **C8** was relative abundant in control reactions (Supplementary Table 4), and it was also detected in model compound reaction in the absence of catalyst. We once considered it was an intermediate with an allene or an alkyne end-chain. However, this possibility could be ruled out as α , β and γ protons in **C4-D** obtained from hydrogenolysis of **M1** in deuterated conditions preserved well. With the prolonging of reaction time, 2,3-dihydro-1*H*-indene-5,6-diol was detected by comparison with the authentic sample, which should be derived from hydrogenation of **C8** (The C=C bond in **C8** adopts *cis*-conformation, which could be readily be hydrogenated.). Thereby, we thought **C8** is a side product rather than the intermediate.

In Figure 3, the structure of **C8** was drawn incorrectly, we have redrawn it.

Some discussions have been added in the manuscript. The synthesis and characterization of 2,3-dihydro-1*H*-indene-5,6-diol and its silylated derivative has been added in the SI.

The authors suggest that C3 is formed via hydrogenation of C4. Why does this not occur in the first part of the reaction? The report 28% C4 after 2 h but only 1% C3 (Table S7). After 4 h they have 51% C4 and 6.6 C3. But then after 6 h they gain more than 38% C3, while the lose only 31% C4. So gain of C3 cannot be explained by loss of C4, at least not completely.

Reply: **C3** could be found in the first part of the reaction with lower yield than **C4** (Supplementary Tables 8 and 9), because the hydrogenation of C=C bonds is far slower than the hydrogenolysis C–O bonds with present Ru/ZnO/C catalyst.

Ru/ZnO/C-catalyzed hydrogenolysis of C-lignin into **C3** should involve the dual C-O bonds cleavage of benzodioxane structure (medium), hydrogenolysis of allylic alcohol of **C1** leading to **C4** and **C5** (fast), and followed hydrogenation (slow). All of these reactions occur at the same time with different reaction rates (see following scheme).

In Table S7 (now it is Table S9), the total monomers yield increased from 66.4% (4 h) to 73% (6 h), because more **C4** and **C5** were released from the cleavage of residual benzodioxane units. Newly generated **C4** and **C5** can also be hydrogenated into **C3**.

They have performed the recycling studies at reaction times of 4 hours where they have already reached their maximum yield and therefore this is not proof of catalyst stability. Again, they should provide proper kinetics or at least do this at lower conversion. They should provide data about Ru and Zn leaching in the liquid phase and recheck the elemental composition of their spent catalysts.

Reply: As suggested, we performed the kinetic experiments by using fresh and spent Ru/ZnO/C (after 1st use), which resulted in comparable kinetic data (as shown below). In view of that it is hard to recover catalyst completely under such a scale, we thought there is no obvious loss of activity in the spent Ru/ZnO/C catalyst. (The spent catalyst was used directly, despite it has adsorbed some organic species).

Reaction condition: C-lignin (180 mg), Ru/ZnO/C (36 mg, 20 wt%), 1,3,5-trimethoxybenzene (20 mg, internal standard) and MeOH (150 mL) at 180 °C under H₂. The spent catalyst was recovered by centrifugation and washed with MeOH, which was then used in the following cycle under same conditions.

Before elemental composition analysis, the spent catalyst (after 1st use) was washed with DMSO and MeOH to remove the organic vestigial. The Ru and Zn contents were determined as 0.18 wt% and 48 wt%, respectively, being slight lower than fresh Ru/Zn/C (0.2 wt% and 51 wt%).

As suggested, we performed elemental analysis of liquid phase and spent catalyst after first run. In the liquid phase, we did not find the Ru and Zn.

We thank this reviewer for the constructive suggestion. These results have been discussed in the manuscript. The kinetic data have been added in the SI.

In conclusion, I believe this work might be publishable but only after major revisions including additional experimental data.

REVIEWER COMMENTS

Reviewer #1 (Remarks to the Author):

Regarding "In this manuscript, we used endocarp of castor seed coats (as shown below), where we did not find the guaiacyl and syringyl-type lignin (β -O-4-type lignin) based on the analysis of 2D NMR and 13C NMR spectra, as well as the analysis of depolymerized products from hydrogenolysis reactions. [...] Previous report has described the coexistence of C-lignin and β -O-4 lignin in some other seed coats (endocarp), which could be identified by 2D NMR spectra (ref. 21)." of the authors' rebuttal:

Ref. 21 does not distinguish between layers of the pericarp in their study of seed coats. Ref. 21 did study castor seed coats (*R. Communis*) finding that they contain both G/S and C lignins. Ref 21 noted the absence of C-lignin only in the stem, root and leaves.

The finding that the endocarp is solely composed of C-lignin is thus novel and significant and warrants more discussion. The authors should provide the reader with the finding that epicarp and endocarp of castor seed coats are differing in composition. In fact, did the authors also use the endocarp in their previous study (10.1021/acssuschemeng.0c00462) and if so, it was an important distinction which should have been specified. The authors should thus include an additional section in their manuscript clarifying this finding and provide the relevant experimental evidence to support it.

Reviewer #2 (Remarks to the Author):

I am satisfied with the authors response to my comments. I recommend publication in its current state.

The point-by-point response to the reviewers' comments

Reviewer #1 (Remarks to the Author):

Regarding "In this manuscript, we used endocarp of castor seed coats (as shown below), where we did not find the guaiacyl and syringyl-type lignin (β -O-4-type lignin) based on the analysis of 2D NMR and ^{13}C NMR spectra, as well as the analysis of depolymerized products from hydrogenolysis reactions. [...] Previous report has described the coexistence of C-lignin and β -O-4 lignin in some other seed coats (endocarp), which could be identified by 2D NMR spectra (ref. 21)." of the authors' rebuttal:

Ref. 21 does not distinguish between layers of the pericarp in their study of seed coats. Ref. 21 did study castor seed coats (*R. Communis*) finding that they contain both G/S and C lignins. Ref 21 noted the absence of C-lignin only in the stem, root and leaves.

The finding that the endocarp is solely composed of C-lignin is thus novel and significant and warrants more discussion. The authors should provide the reader with the finding that epicarp and endocarp of castor seed coats are differing in composition. In fact, did the authors also use the endocarp in their previous study (10.1021/acssuschemeng.0c00462) and if so, it was an important distinction which should have been specified.

The authors should thus include an additional section in their manuscript clarifying this finding and provide the relevant experimental evidence to support it.

Reply: We really appreciate the comments and suggestions from this reviewer. To further explore the structure of C-lignin from castor seed coats, we performed thioacidolysis as suggested in previous comment (Fig. 2). 2D NMR spectra of isolated epicarp lignin (EMAL) was also provided (Fig 1).

For epicarp lignin from castor, 2D HSQC NMR analysis clearly revealed that the epicarp lignin presented a typical herbaceous lignin structure, where guaiacyl (G), syringyl (S) and *p*-coumaric acid (*p*CA) moieties, as well as β -O-4 linkages were detected (Fig 1.). Thioacidolysis indicated that the lignin was composed of G and S units without any trace of C units (Figs 2a and 2c).

Fig 1. 2D NMR spectrum of isolated epicarp lignin (EMAL)

Fig. 2. GC-MS spectra from thioacidolysis analysis

In raw endocarp of castor seed coats, thioacidolysis analysis suggested C units were the main ingredient, where G and S units were also observed (Fig. 2b). **The referee's point in previous comment, i.e. "Castor seed shells contain guaiacyl and syringyl-type lignin" is right.** In isolated C-lignin sample obtained through the combination of enzymatic and mild acidolysis treatment, thioacidolysis only released caffeyl alcohol monomers with essentially no G and S monomers (Fig. 2d), in line with the analysis of 2D NMR and ^{13}C NMR spectra, where no signals from $\beta\text{-O-4}$ structures were detected. Apparently, G and S units were removed during isolation procedure. We proposed it should be occurred during mild acidolysis treatment process, where C-lignin was precipitated and G/S lignin may still remain in the solution.

Of note, the real C-lignin level in raw endocarp of castor seed coats should be high than the value estimated from the released thioacidolysis monomers, because C-lignin biopolymer comprised of benzodioxanes are substantially resistant to thioacidolysis treatments. (ref. 21)

The description on the structure of lignin derived from epicarp and endocarp of castor seed coats have been added in the manuscript and Supplementary Information.

Finally, we thank this reviewer again for these forethoughtful suggestions and comments.